# Exhaustive Plant Profile of *"Dimocarpus longan* Lour" with Significant Phytomedicinal Properties: A Literature Based-Review

Priyanka Paul [1], Partha Biswas [2,3], Dipta Dey [1], Abu Saim Mohammad Saikat [1], Md. Aminul Islam [2], Md Sohel [4], Rajib Hossain [5], Abdullah Al Mamun [4], Md. Ataur Rahman [6,7,8,*], Md. Nazmul Hasan [9,*] and Bonglee Kim [7,8,*]

1 Department of Biochemistry and Molecular Biology, Life Science Faculty, Bangabandhu Sheikh Mujibur Rahman Science and Technology University, Gopalgonj 8100, Bangladesh; paul.bmb011@gmail.com (P.P.); diptadey727@gmail.com (D.D.); asmsaikat.bmb@gmail.com (A.S.M.S.)
2 Department of Genetic Engineering and Biotechnology, Faculty of Biological Science and Technology, Jashore University of Science and Technology (JUST), Jashore 7408, Bangladesh; partha_160626@just.edu.bd (P.B.); aminul_180603@just.edu.bd (M.A.I.)
3 ABEx Bio-Research Center, East Azampur, Dhaka 1230, Bangladesh
4 Department of Biochemistry and Molecular Biology, Faculty of Life Science, Mawlana Bhashani Science and Technology University, Santosh, Tangail 1902, Bangladesh; bmb15021@mbstu.ac.bd (M.S.); amamun42@gmail.com (A.A.M.)
5 Department of Pharmacy, Life Science Faculty, Bangabandhu Sheikh Mujibur Rahman Science and Technology University, Gopalgonj 8100, Bangladesh; rajibhossainrh021@gmail.com
6 Global Biotechnology & Biomedical Research Network (GBBRN), Department of Biotechnology and Genetic Engineering, Faculty of Biological Sciences, Islamic University, Kushtia 7003, Bangladesh
7 Department of Pathology, College of Korean Medicine, Kyung Hee University, Seoul 02447, Korea
8 Korean Medicine-Based Drug Repositioning Cancer Research Center, College of Korean Medicine, Kyung Hee University, Seoul 02447, Korea
9 Laboratory of Pharmaceutical Biotechnology and Bioinformatics, Department of Genetic Engineering and Biotechnology, Jashore University of Science & Technology, Jashore 7408, Bangladesh
* Correspondence: ataur1981rahman@hotmail.com (M.A.R.); mnhasan1978@gmail.com (M.N.H.); bongleekim@khu.ac.kr (B.K.)

**Abstract:** Background: *"Dimocarpus longan* Lour" is a tropical and subtropical evergreen tree species mainly found in China, India, and Thailand; this plant, found naturally in Bangladesh, even locally, is used as "kaviraj" medication for treating different diseases, such as gastrointestinal disorders, wounds, fever, snake bites, menstrual problem, chickenpox, bone fractures, neurological disorders, and reproductive health. Different parts of this plant, especially juice pulp, pericarp, seeds, leaves, and flowers, contain a diverse group of botanical phytocompounds, and nutrient components which are directly related to alleviating numerous diseases. This literature-based review provides the most up-to-date data on the ethnomedicinal usages, phytochemical profiling, and bio-pharmacological effects of *D. longan* Lour based on published scientific articles. Methodology: A literature-based review was conducted by collecting information from various published papers in reputable journals and cited organizations. ChemDraw, a commercial software package, used to draw the chemical structure of the phytochemicals. Results: Various phytochemicals such as flavonoids, tannins, and polyphenols were collected from the various sections of the plant, and other compounds like vitamins and minerals were also obtained from this plant. As a treating agent, this plant displayed many biologicals activities, such as anti-proliferative, antioxidant, anti-cancer, anti-tyrosinase, radical scavenging activity, anti-inflammatory activity, anti-microbial, activation of osteoblast differentiation, anti-fungal, immunomodulatory, probiotic, anti-aging, anti-diabetic, obesity, neurological issues, and suppressive effect on macrophages cells. Different plant parts have displayed better activity in different disease conditions. Still, the compounds, such as gallic acid, ellagic acid, corilagin acid, quercetin, 4-*O*-methyl gallic acid, and (-)-epicatechin showed better activity in the biological system. Gallic acid, corilagin, and ellagic acid strongly exhibited anti-cancer activity in the HepG2, A549, and SGC 7901 cancer cell lines. Additionally, 4-*O*-methyl gallic acid and (-)-epicatechin have displayed outstanding antioxidant activity as well as anti-cancer activity. Conclusion: This plant species can be

considered an alternative source of medication for some diseases as it contains a potential group of chemical constituents.

**Keywords:** *Dimocarpus longan* Lour; immunomodulatory; obesity; neurological disorder; flavonoid; antiproliferative; anti-colorectal cancer; 4-*O*-methyl gallic acid

## 1. Introduction

The Chinese word "*longan*", meaning "*dragon-eye*", conveys an accurate description of the fruit details after removing the fruit's skin [1]. *Dimocarpus longan* Lour, or simply *longan*, is the well-known tropical and subtropical tree species of the Sapindaceae family under the *Dimocarpus* genus growing mainly in South Asian countries. However, China, Thailand, India, and more recently Vietnam, cultivated this plant only for commercial purposes [2]. As well as developing Asian countries, the tree can be found in Central and South American countries, southern African countries, and Australia [3]. However, depending on the climate and soil conditions, the evergreen *longan* tree is approximately 20 m in height, has mild green leaves, unisexual/bisexual flowers, and heart-shaped fruits, i.e., 22–36 mm in diameter and weight of 6–19 g [2]. Here, the edible fresh fruits that have outstanding importance like thin pericarp, soft pulp covering the seed, and hence, the aril is sweeter; of late, the whole fruit has diverse nutrient components that are directly related to medicinal usages [4]. From ancient times, various parts of this plant as pulp, pericarp, seed, leaves, and flowers have health benefits due to it containing fundamental bioactive components, which protect the body from different disorders, namely, insomnia, amnesia, nerve pain, fever, snake bites, gastrointestinal disorders, cuts and wounds, and menstrual problems [5–7].

Generally, *Dimocarpus longan* is a plentiful source of excellent botanical compounds from different parts (pulp, pericarp, seed, leaves, flowers); among them, the pulp is the rudimentary source of nutritional value with excellent ions (i.e., K, Mg, P, Fe, Ca), and most of them have diverse biological functions for human health [8]. Some research output data provided the active amino acid contents within the pulp part of the *longan* fruit juice [9,10]. However, several research findings reported that bioactive phytochemicals from the pulp part of the Lour are phenolic as well as saccharides, both poly- and mono-saccharides [11–15]. Additionally, in recent years, various scientific reports validated that few potential health effective compounds were extracted from the pericarp [16–22]. However, their brownish black seeds are also major sources of botanical phytocompounds [18,20,23–26], furthermore, leaves and flowers produce the major botanical phytoconstituents (ethyl gallate, astragalin, luteolin, gentisic acid, epicatechin, proanthocyanidin) that have potential health benefits and most interestingly, all of them belong to either the polyphenol, flavonoid, or both, groups [27,28].

Aforementioned botanical bioactive compounds have found numerous biological activity by in vivo and in vitro model analysis, for example, *longan* leaf extracts have antiproliferative activity against cancer cell lines, the pericarp extracts including 4-*O*-methylgallic acid and (-)-epicatechin also have potent antioxidant capability and provide health benefits [21]. Additionally, polysaccharides derived from the pulp of the *D. longan* plant effectively affect hepatoma cells (one kind of cancer cell) and must be followed in a dose-dependent manner [29]. Furthermore, the glucans (1-3)-β-D-glucan and (1-6)-α-D-glucan have potent anti-cancer activity, as the experiment conducted by the Iteku Bekomo Jeff and colleagues [30] showed. It is important to mention that extracted *longan* pulp polysaccharides (LP I–IV) directly inhibited the proliferation of HeLa, A549, and HepG2 cancer cell lines at different concentrations ranging from 5.6 to 16.8 percent, 8.3 to 23.2 percent, and 4.7 to 29.5 percent, respectively, and most importantly, LP III inhibited the A549 and HepG2 cells more strongly than other pulp crude extracts [31]. Moreover, three phenolic compounds (gallic acid, corilagin, and ellagic acid) exhibited significant

anti-cancer activity in the SGC 7901, HepG2, as well as A549 cancer cell lines [19]; at the same time, flower and seed extracts of the *longan* plant possessed strong anti-cancer potential on several cancer cell lines via mediating the cancer modulatory pathways [32]. Consequently, inflammation and inflammation-mediated diseases are minimized via mediating the $H_2O$ extract of *longan* pericarp [33]. *Longan* leaf extract and specific extracted chemical components possess significant activity towards the HCV (hepatitis-C virus) and influenza virus infection, respectively [34,35]; lipopolysaccharide (LP-1, 2) derived from the *longan* pulp possesses immunomodulatory activity [12]. Here, ellagic acid showed the most potent anti-fungal activity, *longan* seeds have better anti-fungal activity against opportunistic yeast, for example, *Candida* species and *Cryptococcus neoformans* [36]. According to the findings of several research studies, the leaf extract has anti-aging properties that are dose-dependent [37,38], on the other hand, research outcomes demonstrated that the extracted compounds from the *longan* fruit and water extract of the *longan* flower have been shown to have strong neuroprotective effect through enhancing the survival of immature neurons [39,40]. The seed extract of the *longan* plant have also shown strong anti-diabetic and anti-hyperglycemic effect in both in vitro and in vivo research models by inhibiting glucosidase activity [41]. Moreover, *longan* polyphenol (quercetin) inhibits tyrosinase activity [42]; *longan* flower water extract directly ameliorates hyperlipidemic effects and obesity following the regulation of SREBP-1c with FAS gene expression molecular mechanisms [43,44]. *D. longan* fruit extract directly involves the activation of Erk-1/2 (extracellular signal regulated kinase-1/2) enzyme-dependent-RUNX-2 (runt related transcription factor-2) factors and initiates the differentiation of osteoblasts along with strong activity towards osteoporosis issues [14].

The current review provides more advanced information on the ethnomedicinal uses, taxonomical details, phytochemical profiling, and pharmacological effects of *D. longan* Lour based on published scientific reports and databases.

## 2. Research Methodology

In this current review, all of the significant data were collected and analyzed, as well as summarized, from diverse areas of the *Dimocarpus longan* plant, including botanical description, ethnomedicinal purposes, bioactive phytoconstituents, pharmacological activities through searching PubMed, Google Scholar, Scopus, Willy online sources, ScienceDirect, ResearchGate, SpringerLink, Web of Science, and several patent offices (as-USPTO, CIPO, WIPO). However, all of the published work on *Dimocarpus longan* was cited in this investigation, which is published in English along with distinct keywords, are used for searching information such as *D. longan* and *D. longan* Lour, botanical description, active phytochemicals of *longan*, scientific classification, anti-cancer, anti-microbial and *D. longan*, anti-inflammation, and *longan* plant parts were used. All references listed in the collected articles were also examined to identify further relevant papers. All chemical compound structures were drawn via the ChemDraw tool.

## 3. Results and Discussion

### 3.1. Scientific Classification of Longan and Geographical Description

Taxonomical details of longan tree here—[9]
Kingdom—Plantae
Division—Angiospermea (flowering plant)
Class—Eudicots
Subclass—Rosidea
Order—Sapindales
Family—Sapindaceae
Genus—*Dimocarpus*
Species—*Dimocarpus longan* Lour

China is the original birthplace of the *longan* tree, but it is widespread in all parts of South Asian countries. All tropical and sub-tropical countries produce this tree, but it is mainly propagated in China and Thailand [45]. It is noteworthy that different countries commercially cultivate this tree, for example, China, Thailand, India, and Vietnam [46]. Counter-wise, Crane et al. (2005) [3] reported that the tree was found in Taiwan, Myanmar, Cambodia, Laos, Australia, Kenya, some Central and South American countries, and southern African countries. In Bangladesh, the district of Barisal is most famous for cultivating the *Dimocarpus longan* Lour and it is known locally as "Kath litchi or Ashphal", which is used as an edible fruit as well as for medicinal purposes (mainly used as an antidote) [47].

### 3.2. Complete Botanical Description of D. longan Tree

The *longan* is a very gracious, vertical, and static tree with 20 m height and diameter depending on climate and soil conditions. The orbicular shape at the top of the tree grows with uneven and mercurial peel [48]. Evergreen leaves of the *longan* tree are dilated with 6–9 leaflets per pair of spare and paripinnate leaves. The leaves are up to 30 cm (12 inches) long and 3.5–5 cm wide with deep margins and stingless tops of the leaves. However, the *longan* tree forms shiny leaves with dark green on the upper sides and on the lower base the leaves have a mild green color. Leaves are usually smooth but now and then they have a woolly texture [49]. A *longan* tree usually forms one shoot per year, but sometimes it produces more than one flurry of the shoot and the tree shoots over summer or autumn. Moreover, flowers are small, just 5–6 petalled and the tree produces both unisexual and bisexual (hermaphroditic) flowers [50]. The petals of these flowers are yellow-brown with the tree bearing flowers towards the end of winter. Besides, the female flower conveys a carpellate ovary; flowers are 4–18 inches (10–45 cm) long, held on the panicle in bunch form [3]. *Dimocarpus longan* fruits are small and drupaceous fruit of 22–36 mm in diameter and weight of 6–19 g. Heart-shaped *longan* fruits contain only one seed, and fruits are mostly yellowish to light brown with mellifluous carriage peel. The edible portion is robust with fettle white-peel; furthermore, 350 fruits may be carried by panicles and flowering to harvest is from 140–190 days [3]. Hence, mature *longan* fruit contains one seed inside the fruit; basically, the seed is orbicular with black or brown color with a rounded white spot that has the appearance of a dragon's eye [51]. Various parts of this part are graphically represented under the Figure 1.

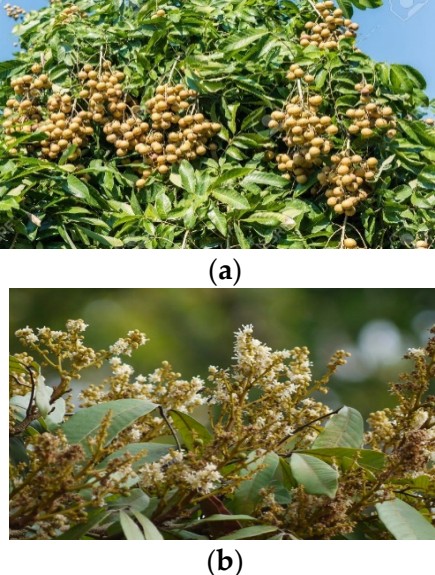

(**a**)

(**b**)

**Figure 1.** *Cont.*

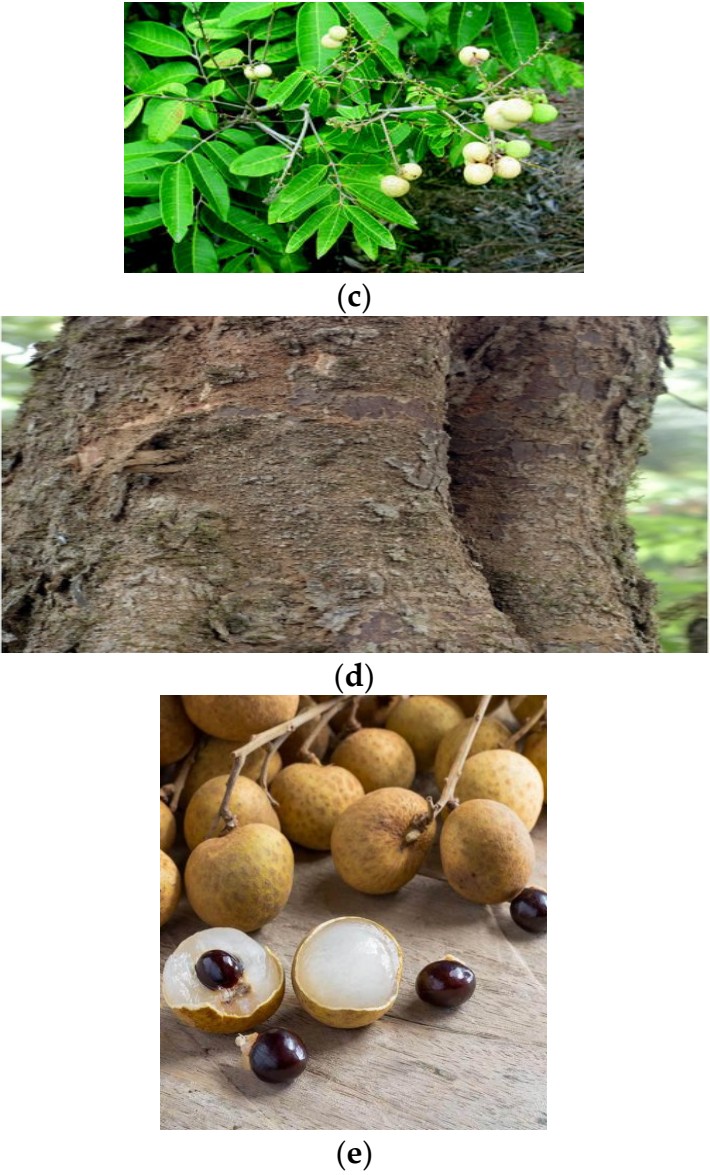

(c)

(d)

(e)

**Figure 1.** Diverse parts of the *D. longan* plant, namely, (**a**) complete fruits, (**b**) flowers, (**c**) leaves, (**d**) trunk, and (**e**) seeds.

### 3.3. Ethnobotanical Usages of Longan

In China, the pulp of the *longan* Lour fruit showed diverse effective health biological functions, for instance, flourishing blood metabolism, calming and relaxing nerves, alleviating insomnia, restraining amnesia, enhancing longevity, relieving nerve pain, curing nerve swelling, and medicating palpitations [14]. Additionally, for a long time, the longan plant was used to treat fatigue diseases. The phytochemical constituents of the flowers and seeds of *longan* decrease the pain associated with urinary disorders. However, the flower, root, pulp, and pericarp have antioxidant, anti-glycation, anti-tyrosinase, anti-fungal, anti-microbial, and anti-cancerous activities. For these reasons, these parts are used in medication for diabetes, cancer, fungal, microbial infections, etc. [14,49]. In Bangladesh, the local "kaviraj" in Barisal use it for different diseases, such as gastrointestinal disorders, cuts and wounds, fever, snake bites, menstrual problems, chickenpox, bone fractures, cattle disorders, and so on; it is more prevalently used as the antidote for poison [47]. In Tabgail, *longan* is used locally to treat neurological disorders and reproductive health [52].

### 3.4. Nutrient Components and Phytochemicals Profiling of Dimocarpus longan

3.4.1. Nutrient Components of the Fruits

Carbohydrates (12–23%), potassium (196.5 mg/100 g), ascorbic acid (43.12–163.7 mg/100 g), and water (about 80%) are all contained in fresh *longan* pulp [8]. Despite not having the maximum polysaccharide content, the fruit pulp is the edible portion widely used in traditional medicine [53]. Fresh *longan* fruit is rich in nutritional components and free amino acids [54]. Dietary compositions and amino acid compositions are illustrated in Tables 1 and 2, respectively. Fresh *longan* fruit pulp contains potassium (266 mg/100 g), which maintains the proper functioning of nerves and muscles of humans [14]. Additional minerals, including iron (Fe), calcium (Ca), phosphorus (P), and magnesium (Mg), are abundant in *longan* fruit pulp. longan fruit pulp is rich in vitamins such as vit-C (ascorbic acid), riboflavin, thiamin, and niacin (Table 1). Furthermore, water, protein, ash, carbohydrate, and fiber are available in the Lour. fruit pulp. Fresh *longan* fruit pulp contains seven essential amino acids, and most importantly, few free amino acids, namely, glutamic acid (Glu), alanine (Ala), aspartic acid (Asp), valine (Val), and leucine (Leu) are found (Table 2).

**Table 1.** Nutritional content per 100 g of fresh *D. longan* Lour fruit pulp (acquired from the USDA National Nutrient Database [14].

| Type | Content | Type | Content |
|---|---|---|---|
| Water | 82.75 g | Total lipid (fat) | 0.1 g |
| Energy | 60 kcal | Calcium (Ca) | 1 mg |
| Protein | 1.31 g | Iron (Fe) | 0.13 mg |
| Ash | 0.7 g | Phosphorus (P) | 21 mg |
| Carbohydrate | 15.14 g | Potassium (K) | 266 mg |
| Fiber (total dietary) | 1.1 g | Thiamin | 0.031 mg |
| Magnesium (Mg) | 10 mg | Niacin | 0.3 mg |
| Vit-C (ascorbic acid) | 84 mg | Riboflavin | 0.14 mg |

**Table 2.** Amino acid (aa) composition per 100 g of fresh *D. longan* Lour fruit pulp (acquired from the USDA National Nutrient Database (https://fdc.nal.usda.gov/fdc-app.html#/food-details/169089/nutrients; accessed on 10 September 2021).

| Type | Content (g) | Type | Content (g) |
|---|---|---|---|
| Threonine (Thr) | 0.034 | Leucine (Leu) | 0.054 |
| Isoleucine (Ile) | 0.026 | Lysine (Lys) | 0.046 |
| Methionine (Met) | 0.013 | Tyrosine (Tyr) | 0.025 |
| Phenylalanine (Phe) | 0.03 | Valine (Val) | 0.058 |
| Arginine (Arg) | 0.035 | Alanine (Ala) | 0.157 |
| Histidine (His) | 0.012 | Glutamic acid (Glu) | 0.209 |
| Glycine (Gly) | 0.042 | Proline (Pro) | 0.042 |
| Serine (Ser) | 0.048 | Aspartic acid (Asp) | 0.126 |

3.4.2. Phytochemical Profiling

A vast amount of potential bioactive phytoconstituents have also been isolated from different parts of the *longan* Lour tree and all compounds are reported within Table 3. Moreover, pulp is the chief source of the major phytocompounds, for instance, protocatechuic acid, vanillic acid, caffeic acid, 4-methylcatechol, *p*-Coumaric acid, ferulic acid, syringic acid, chlorogenic acid, quinic acid, narirutin, naringin, rhoifolin, hesperidin, phthalic acid, methyl hesperidin, naringenin, phlorizin, gallic acid, epicatechin, (-)-epicatechin, isoquercitrin, and coumarin [11,14,15,55]. On the contrary, pericarp accommodates many compounds, including protocatechuic acid, ellagic acid, ethyl gallate, gallic acid, corilagin, isoscopoletin, brevifolin, 4-*O*-methylgallic acid, proanthocyanidin trimer (a type), (-)-epicatechin, quercetin, proanthocyanidins c1, methyl gallate, methyl brevifolin carboxylate, and rutin [16–22]. Furthermore, the *D. longan* seed contains corilagin, gallic acid, ellagic acid, 3′-o-methyl-ellagic acid 4′-o-β-d glucopyranoside, ethyl gallate, geraniin,

(s)-flavogallonic acid, as well as isomallotinic acid [18,20,23–26]. Furthermore, diverse research findings indicated that the leaves and flowers of *longan* Lour consist of several botanical phytochemicals, namely, ethyl gallate, astragalin, luteolin, kaempferol, quercetin, gentisic acid, epicatechin, and proanthocyanidin [27,28]; of note, most of the chemical compounds belong to either the polyphenol, flavonoid, or both groups. The chemical structure of all compounds is represented in Figure 2.

**Table 3.** Tabular representation of extracted bioactive phytochemicals from the different parts of the *D. longan* Lour plant.

| Plant Parts | Phytocompounds | References |
|---|---|---|
| Pulp | Protocatechuic acid, Vanillic acid, 4-Methylcatechol, *p*-Coumaric acid, Ferulic acid, Syringic acid, Chlorogenic acid, Quinic acid, Caffeic acid, Narirutin, Naringin, Rhoifolin, Hesperidin, Phthalic acid, Methyl hesperidin, Naringenin, Phlorizin, Gallic acid, Epicatechin, Isoquercitrin, coumarin | [11,14,15,55] |
| Pericarp | Protocatechuic acid, Ellagic acid, Ethyl gallate, Gallic acid, Corilagin, Isoscopoletin, Brevifolin, 4-*O*-methylgallic acid, Proanthocyanidin, Epicatechin, Quercetin, Proanthocyanidins C1, Methyl gallate, Methyl brevifolin carboxylate, Rutin | [16–22] |
| Seeds | Corilagin, Gallic acid, Ellagic acid, Ethyl gallate, Geraniin, Flavogallonic acid | [18,20,23–26] |
| Leaves | Ethyl gallate, Astragalin, Luteolin, kaempferol, Quercetin | [27] |
| Flowers | Gentisic acid, Epicatechin, Proanthocyanidin | [28] |

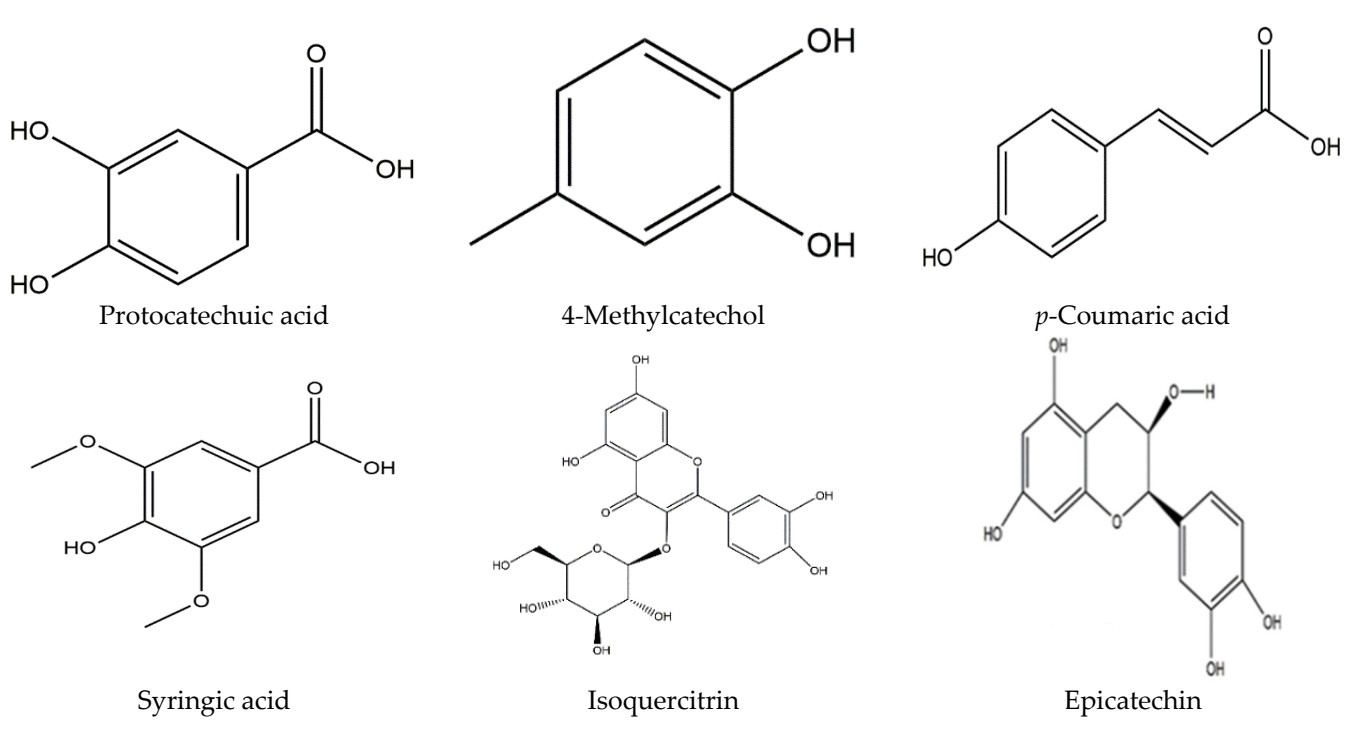

Protocatechuic acid     4-Methylcatechol     *p*-Coumaric acid

Syringic acid     Isoquercitrin     Epicatechin

**Figure 2.** *Cont.*

Chlorogenic acid

Ferulic acid

Ellagic acid

Caffeic acid

Quinic acid

Narirutin

Naringin

Rhoifolin

Hesperidin

Phthalic acid

Naringenin

Methyl hesperidin

Gallic acid

Kaempferol

Phlorizin

**Figure 2.** *Cont.*

Corilagin

Proanthocyanidins C1

Quercetin

Ethyl gallate

Vanillic acid

Brevifolin

4-*O*-methylgallic acid

Rutin

Geraniin

Flavogallonic acid

Isoscopoletin

Proanthocyanidin

Methyl brevifolin carboxylate

Astragalin

Luteolin

**Figure 2.** *Cont.*

|　Gentisic acid　|　Coumarin　|　Methyl gallate　|

**Figure 2.** Diagrammatic representation of different chemical structures.

### 3.5. Pharmacological Activities of Dimocarpus longan

The pharmacological investigation of the *D. longan* Lour are as follows and the data are summarized in Table 4.

### 3.5.1. Antiproliferative, Antioxidant Activity and Anticancer Activity

The excessive, abnormal, and uncontrolled growth of the body's tissue cells are characteristics of cancer. Cancer cells (invasive) infiltrate and continue to expand the surrounding tissue (metastasis). Secondary metabolites are plant-derived compounds with bioactivity that can inhibit cancer cell proliferation [56]. The antiproliferative activity of *D. longan* leaf extracts against cell lines derived from cancer was studied in a controlled environment and in in vivo research models. The research study also established that *longan* leaf ethanol extract possessed significant antiproliferative activity against cancer-derived cell lines. Table 4 also shows the significant pharmacological activities of isolated compounds of *longan*. The highest antiproliferative activity was obtained by extracting WEHI-164 at 600 μg/mL and 57.45 percent by 500 μg/mL of ethanol at THP-1 at 44.93 percent [57].

Antioxidants are the chemical substances that can improve shelf-life by delaying the oxidation process when incorporated into cellular components, namely, DNA/RNA, protein, and lipid molecules, which are one of the main reasons for foodstuff degradation during production and storage [58]. Accordingly, bioactive compounds, particularly from plant sources, have become more critical in recent years [59]. Many plant-derived bioactive compounds, and crude vegetable and fruit extracts were known to positively affect the free radicals in biological systems as significant antioxidant compounds [60,61]. The pericarp of the longan fruit is densely packed with bioactive substances such as phenolic compounds, polyphenols, hydrolyzable tannins, and polysaccharides. Those compounds had considerable antioxidant activity in different models of antioxidants, including 1,1-diphenyl-2-picrylhydrazyl (DPPH) radical scavenging, radical scavenging activity of superoxide anions, total antioxidative capacity, and inhibitory lipid peroxidation activity [22]. Phenolic compounds of *longan* plant parts have long been thought to possess significant antioxidant and free radical scavenging properties, due to its ability to suppress the enzymes responsible for the production of reactive oxygen species (ROS) and to reduce rapidly oxidized ROS [62,63]. To further investigate the findings of various research, Fu et al. discovered that *longan* possessed a ferric reducing antioxidant power (FRAP) value of $8.61 \pm 0.44$ μmol Fe (II)/g and a total phenolic value of $5.88 \pm 0.34$ μmol Trolox/g. The study also revealed a strong interaction (R2 = 0.8416) among the FRAP value and total phenolic content [59,64]. Several crops of *longan* have been studied for their antioxidant potential, and the cellular antioxidant activity (CAA) scores ranged from 0.49 to 6.71 mol quercetin equivalents (QE)/100 g of fruit with an average value of 2.76 mol QE/100 g of fruit. According to CAA values, the antioxidant activity of *longan* fruit appears to be dominated by phenolics and flavonoids [11]. The FRAP value of longan plant seed was also greater compared to the *longan* peel and pulp [65], in which the pulp has the lowest FRAP value within the three components. In addition, 4-*O*-methyl gallic acid and (-)-epicatechin also have antioxidant capabilities and health benefits extracted from the pericarp [22].

It is now known that cancer is the number one health threat to the general population, and thus we need to prevent and treat it by using potential strategies [66]. It is increasingly a preventable disease because cancer develops progressively slowly and takes many years to become a life-threatening condition [67–69]. A pure polysaccharide (LPS1) derived from the pulp of the *longan* plant has a dose-dependent manner for the significant effect on hepatoma cells, most likely due to the immunomodulatory activities of (1–6)-α-D-glucan [29]. The research study by Iteku Bekomo Jeff and colleagues demonstrated that the anti-cancer activity of the glucans (1-3)-β-D-glucan and (1–6)-α-D-glucan were confirmed [30]. In the in vitro studies, a new water-soluble polysaccharide derived from the *longan* pulp (LP1) demonstrated a significant anti-tumor effect on the SKOV3 and HO8910 cancer cell lines, with the antiproliferative percentages of 40 percent at a concentration of 40 mg/L and 50 percent at a concentration of 320 mg/L, respectively, at different concentrations [12]. Four extracted *longan* polysaccharides (LP I–IV) and refined longan pulp polysaccharides inhibited the proliferation of A549, HeLa, and HepG2 cancer cell lines at different concentrations ranging from 5.6 to 16.8 percent, 8.3 to 23.2 percent, and 4.7 to 29.5 percent, respectively, and LP III inhibited A549 and HepG2 cells more strongly than refined or crude *longan* pulp polysaccharides [31]. The insights of the factors associated with polysaccharide anti-tumor function were in the following order: water solubility > chain conformation > average molar masses (Mw) [70]. According to cancer epidemiological studies, enhancing the consumption rate of phenolic contents is associated with a lower risk of cancer formation [71–74]. Three phenolic compounds exhibited significant anti-cancer activity in the HepG2, A549, and SGC 7901 cancer cell lines: gallic acid, corilagin, and ellagic acid [19]. Chih-Cheng Lin et al. (2012) [32] noted that the extracted compounds of *longan* flower and seed possessed strong anti-cancer potential on several cancer cell lines through inhibiting the cancer modulatory pathways.

### 3.5.2. Anti-Inflammatory Properties

Inflammation has been defined as the tissue's localized protective response to injury or infection, manifested by pain, redness, and swelling. The inflammatory process involves several physiological systems with a central role in the immune system. Several molecules and signaling pathways are upregulated in damaged areas as a result of inflammation. The inducing features of nitric oxide synthase (iNOS) and cyclooxygenase-2 are these pro-inflammatory enzymes (COX-2). Increased levels of nitric oxide (NO) and prostaglandins (PGs) are caused by the genes iNOS and COX-2, respectively [75]. Most strong evidence for NO's role as a mediating role of the inflammatory response has come from studies on an animal rheumatoid model, human osteoarthritis, and rheumatoid arthritis, among other sources [76].

Additionally, to cope with the increase in oxidative stress and inflammation that occur during injury, tissues contain antioxidant enzymes such as superoxide dismutase (SOD), catalase (CAT), and glutathione peroxidase (GPx). In recent times, it was demonstrated that dysfunctional cellular antioxidant mechanisms contribute to the development of a number of adverse and cancerous diseases in organisms [77]. There is evidence that the critical roles played by antioxidant enzymes in the inflammation pathway defend the organisms from oxidative stress [78]. The suppression of NO and tumor necrosis factor (TNF) as well as the enhancement of antioxidant enzyme activities, such as catalase, superoxide dismutase, and glutathione peroxidase, have shown that the water extract of *longan* pericarp (WLP) has anti-inflammatory properties [33].

### 3.5.3. Immunomodulatory Activities

Polysaccharides derived from a variety of natural sources have been shown to possess immunomodulating properties [14,79]. LPD2, an effective polysaccharide derived from *longan* pulp, demonstrated a significant effect on the upregulation of macrophages phagocytic effect as the multiplication of splenic lymphocytes through the toll-like receptor 2 (TLR2) and 4 (TLR4) facilitated myeloid differentiation factor 88/interleukin receptor-

associated kinases (MDF88/ILRK) signaling pathway and the tumor necrosis factor receptor-associated factor 6 (TRAF6) signaling pathway [12,80–82]. The major reason why LPD2 is the stronger immunomodulatory substance is higher molecular weight, acetyl groups, and (1–4)-β-Glc. LP1 and LP1-S were shown to significantly raise the pinocytic effect of murine macrophages and development of nitric oxide (NO), interleukin 6 (IL 6), interleukin (IL-1), and tumor necrosis factor-alpha (TNF-alpha) in vitro, according to experimental research [83]. Cytokines released during the immune response by the helper T-lymphocyte play a significant role in controlling the existence of the reaction. For instance, type 1 helper T-cells (Th1 cells) release interferon (IFN- γ) and interleukin-2 (IL-2) to modulate cell-mediated immunity [84]. IFN- is a multifunctional cytokine that has immunomodulatory effects on a variety of immune cells. IFN- has been shown in mammals as a marker of cellular immunity in infected organisms [85]. Consequently, the IFN-$\alpha$ detection can be made to preliminarily evaluate T cell activation's extent [86]. The water-soluble polysaccharide (LP1) extracted from *Dimocarpus longan* pulp has shown solid immunomodulatory activities. The research studies have significantly demonstrated that the LP1 have effectively regulated the expression of the cytokine interferon-γ (IFN-γ) and enhanced the activity of murine macrophages and the B- and T-lymphocyte production [12].

### 3.5.4. Prebiotic Activities

Prebiotics are the functional foodstuffs categorized as edible products that have to be measured by their health benefit by their intake in the bloodstream, and by the component's main activity [87]. The non-digestible carbon-hydrates, such as resistant starch, galacto-oligosaccharides (GOS), fructo-oligosaccharides (FOS), and various oligosaccharides that produce carbohydrates fermentable by advantageous colon microorganisms are prebiotics that are obtained from natural sources such as vegetables, rootstock, fruit, milk, or honey [88,89]. The research studies noted that the *longan* pulp polysaccharides showed intense prebiotic activity on several probiotic bacterial strains. The superfine grinding-assisted enzymatic treatments (LP-SE) of longan pulp polysaccharides exhibited the most important prebiotic activities with great potential in the use of functional food and medical industries [90]. The polysaccharides from the pulp of *longan* had more significant effects on *Lactobacillus plantarum*, *Lactobacillus bulgaricus*, *Lactobacillus fermentum*, and *Leuconostoc mesenteroides* than LP-H (*longan* pulp polysaccharide extracted using warm water) and LP-S (*longan* pulp polysaccharide extracted using superfine grinding) [90]. *Longan* cellulose with a degree of hydrolysis of 21% demonstrated a greater prebiotic significance and growth level of bacteria for *Lactobacillus acidophilus* and *Bifidobacterium lactis* [91].

### 3.5.5. Anti-Microbial Activities

Plant extracts and phytochemicals, both of which have been shown to have anti-microbial properties, can become very useful in therapeutic approaches [92]. Several studies in various countries have been carried out to demonstrate this efficiency [93]. Due to the production of compounds in the plant secondary metabolic pathways, several species of plants have been used for their anti-microbial properties. These products are characterized by essential ingredients, such as phenolic content found in essential oils known as tannin [94,95]. The anti-microbial properties of *longan* Lour seed extracts were examined using disc diffusion methods, and the minimum inhibitory concentration was determined. The DL-P01-SI01 (*Dimocarpus longan*: crude methanolic extract; fractions: DL-P01, aquation; ethyl acetate subfractions) fraction demonstrated the highest activity against *Staphylococcus aureus* and methicillin-resistant *S. aureus* at an MIC of 64 mg/mL, attributed to the phenolic compounds [24]. Apriyanto et al. (2015) [34] reported that the *longan* tree leaf extract possesses activity towards the hepatitis-C virus and minimizing of death rate. Anti-influenza activity has also been noted by the chemical components from the parasitic plant on *Dimocarpus longan* Lour [35].

### 3.5.6. Anti-Fungal Activities

Intermittent fungi cause severe disease and mortality in patients with weak immune conditions [96]. *Candida* can be found in the normal flora of the mouth, skin, intestines, and vaginal area. *Candida albicans* is one of the *Candida* species found in the oral cavity and is responsible for most oral candidal infections [97]. *Cryptococcus neoformans* is a yeast-like encapsulated fungus that causes central nervous system and pulmonary problems in immunocompromised people and is an opportunistic fungal infection in both plants and animals [98,99]. The results of many studies have shown that *longan* seeds have anti-fungal activity against opportunistic yeast (*Candida* species and *Cryptococcus neoformans*). Ellagic acid showed the most potent anti-fungal activity, followed by corilagin and gallic acid, respectively, from all the extracted *longan* compounds. *Candida krusei* and some *Candida albicans* clinical strains were more efficiently suppressed by ellagic acid than *Candida parapsilosis* and *Candida neoformans* [36]. The significant pharmacological activities of isolated compounds of *longan* are represented in Table 4.

### 3.5.7. Neuroprotective Activities

Human brain synaptic vesicles contain the neurotrophin, brain-derived neurotrophic factor (BDNF), composed and deposited in the synaptic lesions to respond to endogenous or exogenous transmissions. Integration with the trkB receptor or the p75NTR receptor reveals its characteristics by interacting with the tropomyosin-related tyrosine kinase receptor B (trkB) or the p75 neurotrophin binding site p75NTR [100,101]. BDNF, which assists in neuronal transmission and memory incorporation, is an important component in producing and maintaining long-term memory synaptic transmission [102,103]. Neurogenesis appears to occur repetitively all throughout adulthood in two areas of the adult brain, known as the subventricular zone (SVZ) of the lateral ventricles and the subgranular zone (SGZ) of the dentate gyrus (DG) of the brainstem [104–106].

Additional findings have shown that brain-derived neurotrophic factor (BDNF) is necessary to preserve neuronal cells throughout development and neurogenesis [107,108]. In general, it seems that BDNF cascades and neurogenesis are a memory development procedure. In recent years, a growing number of studies have focused on neuroprotective strategies involving dietary supplements for the therapeutic interventions of central nervous system neurodegenerative disorders [109–111]. In the mice research model, the extracted compounds from the longan fruit part have shown a strong neuroprotective effect through enhancing the survival of immature neurons [39]. The research study by Anya Maan-Yuh Lin and colleagues reported that the water extract of *longan* flower possessed a potent neuroprotective effect in the brain rat model developed with the MPP$^+$-induced neurotoxicity [40].

### 3.5.8. Anti-Aging Activities

Ageing is characterized by progressive disintegration of cells, a significant risk factor for developing a wide variety of degenerative diseases, including cardiovascular disease, neurodegenerative disease [112], and even skin ageing [113]. Many other research studies also reported that the phytochemical component of *longan* leaves showed potential anti-ageing characteristics. The *longan* leaves hydroethanolic extract (HE) demonstrated radical activity in the experimentation of DPPH and hydrogen peroxide with $IC_{50}$ values of $30.03 \pm 7.64$ and $71.40 \pm 15.30$ μg/mL, respectively. Moreover, it showed inhibition of lipid peroxidation with $IC_{50}$ of $537.01 \pm 42.32$ μg/mL. The HE was found to inhibit hyaluronidase and collagenase with $IC_{50}$ of 234.80, 21.52 and 314.44 62.14 g/mL, respectively. The extract also showed inhibition of MMP-2 and MMP-9 that is more potent than gallic acid by zymography at 1.0 mg/mL [38].

### 3.5.9. Anti-Diabetic Effect and Anti-Hyper Glycemic Effects

Diabetes mellitus (DM) was one of the world's leading causes of death. This figure is expected to reach 438 million by 2030 when misdiagnosed cases of diabetes are also included [114]. Subsequent studies have shown that hyperlipidemia and oxidative stress each play an important role in developing diabetes, with each increasing the risk of abnormalities [115]. As a result, there are many oral hypoglycemic medication therapies for the management of diabetes, such as biguanides and sulfonylureas, but these medications can produce severe side effects [116]. The research study by Ya-Yuan Tang and colleagues reported that the polyphenols and alkaloids from extracted by-products of the *longan* fruits possessed a strong anti-diabetic effect in vitro [18]. The pericarp extract of the *longan* plant revealed the potent anti-diabetic with anti-hyperglycemic activity in the mouse model by enhancing the gene expression associated with the production of insulin [17]. Moreover, the seed extract also showed strong anti-diabetic and anti-hyperglycemic effect on both the in vitro and in vivo research models by inhibiting the glucosidase activity [41].

### 3.5.10. Anti-Tyrosinase Properties

Browning of crude fruits, vegetables and beverages is a serious problem in the food processing industry and one of the major causes of postharvest quality loss during collection and management [117,118]. Browning of fruits and vegetables due to enzymatic action is primarily due to the oxidation of endogenous phenolic compounds [119]. The phenolic oxidation is known to be caused by an enzyme that is known as tyrosinase (monophenol, o-diphenol: oxygen oxidoreductase; EC 1.14.18.1). It is widespread in microorganisms, animals, and plants and is also responsible not only for plant browning but also animal melanization [120].

It has been demonstrated that the *longan* pericarp extract has anti-tyrosinase activity. When looking at ultra-high-pressure-induced extraction of 500 MPa and traditional extractions, the *longan* pericarp extract from the ultra-high-pressure-induced extraction exhibited the greatest proportion of anti-tyrosinase property, $23.6 \pm 1.2\%$ at the concentration of 100 g/mL, when compared to traditional extraction [121]. The mechanism of action of some tyrosinase inhibitors is via hydrophilic groups that attach with the active site of an enzyme, causing steric hindrance or altered conformation [122]. According to the study by Rout and Banerjee, the ultrasonication of polysaccharides from *longan* fruit pericarp (PLFP) inhibited tyrosinase activity non-competitively [123,124]. A wide variety of tests conducted on fresh and processed *longan* seed extracts revealed tyrosinase inhibition, and the $IC_{50}$ values for fresh and processed extracts were 2.9 and 3.2 mg/mL, comparatively [23]. The polyphenols of *longan* have also been shown to have tyrosinase inhibitory activity. In their study, Guan et al. discovered that the inhibitory effect of *longan* polyphenol extract on tyrosinase activity was dose-dependent. The inhibitory impact also resulted in a high sample concentration rate. It is conceivable that ellagic acid, gallic acid, corilagin, and ethyl gallate are responsible for inhibitory activity. Such compound holds various hydroxyl groups that are structurally similar to the substrate and have the potential to attach to the copper ion active site of tyrosinase, removing active oxygen and inhibiting tyrosinase enzyme expression [14]. The results of inhibitory activity studies confirmed that *longan* polyphenols inhibited tyrosine diphenolase in a reversible and competitive manner. As a result, the combined effect of *longan* polyphenols and substances with enzymes does not create an irreversible change in the cognitive shape of the enzyme. It uses *longan* polyphenols as a highly competitive aid to copper ions that inhibits tyrosinase formulation in the catalyzed reaction, which ultimately reduces the level of tyrosinase in the reaction mixture. Kubo et al. also characterized the methodology by which quercetin inhibits tyrosinase activity; they discovered that quercetin inhibits tyrosinase activity compared to active tyrosinase centers [42].

### 3.5.11. Miscellaneous Activities

Obesity is regarded as serious health condition that leads to the manifestation of diverse health problems, including cardiovascular disease, diabetes mellitus, hypertension, fatty liver, some cancers, mental health problems, and so on, and thereby this is a life-threatening problem [125], as these conditions increase the lipid levels of our bodies. Here, the *longan* flower water extract directly ameliorates the hyperlipidemic effects and obesity with such effective activity showed by the polyphenol compounds. The following mechanisms act as control of the expression level of hepatic PPAR-alpha gene, regulation of SREBP-1c with FAS gene expression, reducing the exogenous lipid absorption, and the large amount of the fecal TG (triglyceride) output. Additionally, the total biological process occurred within the in vitro rat model in a dose-dependent manner [43,44].

A study by Zhu et al., 2016 [13], demonstrated that the polysaccharides of *D. longan* pulp could significantly promote the upregulation of sox9, aggrecan, and collagen II gene expression, consequently, synthesis of the CAM (cartilage extracellular matrix) protein as well as chondrocyte act as an excellent activity towards osteoporosis; the experiment was conducted within an in vitro model animal. On the other hand, *D. longan* fruit extract directly involves the activation of Erk-1/2 (extracellular signal regulated kinase-1/2) enzyme-dependent-RUNX-2 (Runt related transcription factor-2) factors via following the phosphorylation mechanism along with initiating the differentiation of osteoblasts. *Longan* fruit extract also represses the mRNA expression of osteoclast and thereby inhibits the differentiation of osteoclast, mediates the osteoporosis disease severity, and decreases the TRAP (tartrate resistant acid phosphatase) protein-mediated multinucleated cells in the RAW264.7 cells [14]. Additionally, NF-κB (nuclear factor-kappa B) pathway downregulation, NFATc1 (nuclear factor of activated T-cells c1) suppression through the *longan* Lour fruit extract efficiently involves the suppression of osteoclast differentiation in vitro. Additionally, the administration of Lour fruit extract in vivo experiment model ovariectomized rats and zebrafish enhanced their mineral contents in bone, minimizing bone disorder risk [126,127]. *Longan* Lour flower $H_2O$ extract attenuates the serological TG (triglyceride), disaggregates the lipid moiety, and downregulates the MMP-2,9 (matrix metalloproteinases-2,9) gene expressions, thereby protecting the hepatic cells; in vitro hypercaloric-dietary rat model study [28].

**Table 4.** Potential pharmacological activities of "*Dimocarpus longan*" plant.

| Potential Activity | Sources | Compound Name and Chemical Class | Test System | Test Dose/Concentration | Results | References |
|---|---|---|---|---|---|---|
| Antioxidant activity | High pressure-assisted extract of fruit pericarp | Gallic acid, corilagin acid, and ellagic acid | Phosphomolybdenum method using various antioxidant model systems | 50 µg/mL at 90 min | Strong antioxidant activity | [19] |
| | Ultrasonic assisted extract of Fruit Pericarp | Galactose and galacturonic acid | OLFP and DPPH Radical Scavenging Assay | At the concentration of 500 µg/mL | Strong antioxidant activities | [128] |
| Anti-tyrosinase activity | Ultra-high-pressure-assisted fruit pericarp extract | Phenolic acids, gallic acid, ellagic acid, and corilagin | Through HPLC assay | l-Tyrosine solution (4 mL) at 0.5 mg/mL, dissolved in 20 mM phosphate buffer (pH 6.8) | Enhanced anti-tyrosinase activity | [121] |
| Anti-glycated activity | Extract of fruit polysaccharides with Ultrasonic wave | Plant polysaccharides, mainly the phenolic compounds | PLFP assay and aminoguanidine | At the concentration of 0.5 mg/mL | Significant anti-glycated activity | [129] |
| Radical-scavenging activity | Polyphenols from seeds | Polyphenols (methyl brevifolin carboxylate, brevifolin and 4-O-a-L-rhamnopyranosyl-ellagic acid) | DPPH radical assay and superoxide radical assay | 0.80– 5.91 lg/mL for DPPH radical assay and 1.04–7.03 lg/mL superoxide radical assay | Effective radical-scavenging activity | [130] |
| Anti-Inflammatory Properties | Water extract of *longan* pericarp | Polyphenols | Male ICR mice (6–8 weeks) | (10 mg/kg) | Strong anti-Inflammatory properties | [33] |
| Anti-microbial activities | Seed extracts | Phenolic compounds (gallic acid, corilagin, ethyl gallate and ellagic acid) | Disc diffusion method | 64 mg/mL | Strong antimicrobial activities | [24] |
| Activation of osteoblast differentiation | Fruit Extract | Plant Polyphenolic Compounds | Promotion of signal-regulated kinase1/2 (Erk1/2) | 500 µg/mL | Can activate osteoblast differentiation | [127] |
| Anti-fungal activities | Seed extract | Ellagic acid, corilagin acid and gallic acid | Disc–agar diffusion assay | 15.63–16,000 µg/mL | Potential antifungal activities | [36] |
| Anti-colorectal cancer effects | The polyphenol of seed extract | Phenolic compounds | CRC cell lines (Colo 320DM, SW480, HT-29 and LoVo) | 25 µg/mL–200 µg/mL | Strong anti-colorectal cancer effects | [131] |
| Antitumor activities | Water extract of pulp | Monosaccharide compounds | SKOV3 and HO8910 tumor cells | 5–40 mg/L | Effective antitumor activities | [12] |

**Table 4.** *Cont.*

| Potential Activity | Sources | Compound Name and Chemical Class | Test System | Test Dose/Concentration | Results | References |
|---|---|---|---|---|---|---|
| Immunomodulatory activities | Water extract of pulp | Monosaccharide compounds | Immunosuppression of serum IL-2 levels in mice | 320 mg/kg | Effective Immunomodulatory activities | [12] |
| Articular chondrocytes maintenance activity | Pulp extract | Plant polysaccharides | Articular chondrocytes culture 1-week-old New Zealand rabbits | 9.38 µg/mL | Intense articular chondrocytes maintenance activity | [13] |
| Prebiotic activities | Pulp extract | Plant polysaccharides | Basal medium | 0.5, 1.0, 1.5 and 2.0% (*w/v*) | Potential prebiotic activities | [87] |
| Flocculant in landfill leachate treatment | Seed powder | Not demonstrated | Landfill leachate samples | 2 g/L LSP and 2.75 g/L PACl | Show effective efficiency | [132] |
| Anti-ageing activities | Leaf Extracts | Plant total phenolic and flavonoid content | MTT assay on mouse embryonic fibroblasts (BCRC 60071; ATCC CCL92) | 0.1–1 mg/mL | Potential anti-aging activities | [38] |
| Suppressive effect on macrophage cells | Flower extract | Flavonoids (tannins, and proanthocyanidins) | Determination PGE2 by enzyme immunoassay | 1 µg/mL | Strong suppressive activity | [133] |

## 4. Concluding Remarks

Nowadays, natural food products are given more attention by people to combat diseases, including cardiovascular diseases, immune dysfunctions, and cancer insurgencies. Additionally, consumers are turning to compounds derived from medicinal plants to treat a wide range of conditions, including malignancy, due to the lower risk of complications and lower cost of these biomolecules. It was recently found that researchers from pharmaceutical sectors and medication are searching for natural compounds as medicinal agents since synthetic compounds show substantial side effects to the patients' bodies. For this reason, this review was conducted to explore natural phytochemicals that offer therapeutic activities; as a model plant, *Dimocarpus longan* Lour was reviewed, and it significantly exhibited a diverse group of chemical compounds. As a source of flavonoid and phenolic components, this plant displayed different biological activities, and more interestingly, it showed strong anti-cancer and anti-diabetic activities.

Consequently, this review article has demonstrated that the compounds derived from *Dimocarpus longan* Lour will be used as a complementary and alternative medicine to treat many different types of diseases. They can also serve as possible sources of phytotherapeutic lead molecules. However, according to previously published research, the pharmacokinetic evidence for this promising, highly nutritious medicinal plant and its derivative products is insufficient in this case. Therefore, more research on these natural compounds is highly required, especially on their toxicogenetical profiles. Therefore, more research is needed to discover the specific disease controlling and toxicological mechanisms and their pharmacokinetics properties.

**Author Contributions:** D.D. and P.B. contributed equally to the conceptualization and study design; P.P., P.B., D.D., A.S.M.S., M.A.I. and R.H. participated equally in data collection, writing, and draft preparation; R.H., M.S. and A.A.M. drew all chemical structures and visualizations; M.A.R., M.N.H. and D.D. performed reviewing and editing; M.A.R., M.N.H. and B.K. visualized and supervised; B.K., funded the project. All authors have read and agreed to the published version of the manuscript.

**Funding:** This research was supported by Basic Science Research Program through the National Research Foundation of Korea (NRF) funded by the Ministry of Education (NRF-2020R1I1A2066868), the National Research Foundation of Korea (NRF) grant funded by the Korea government (MSIT) (No. 2020R1A5A2019413), a grant from the Korea Health Technology R&D Project through the Korea Health Industry Development Institute (KHIDI), funded by the Ministry of Health & Welfare, Republic of Korea (grant number: HF20C0116), and a grant from the Korea Health Technology R&D Project through the Korea Health Industry Development Institute (KHIDI), funded by the Ministry of Health & Welfare, Republic of Korea (grant number: HF20C0038).

**Conflicts of Interest:** The authors declare no conflict of interest.

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
