# Peer review of "Exhaustive Plant Profile of “Dimocarpus longan Lour” with Significant Phytomedicinal Properties: A Literature Based-Review"

_processes, doi:10.3390/pr9101803_

Round 1

Reviewer 1 Report

As far as the observations I made at paper are concerned, they are accepted by the authors. I would advise the authors to carefully double-check throughout the manuscript if “Dimocarpus longan Lour” is written appropriately and likewise and to carefully double-check the formatting of the references.

Author Response

As far as the observations I made at paper are concerned, they are accepted by the authors. I would advise the authors to carefully double-check throughout the manuscript if “Dimocarpus longan Lour” is written appropriately and likewise and to carefully double-check the formatting of the references

>>Response: Firstly, thanks to your suggestions. We have checked the total manuscript and updated the information marked as blue color. Also, we have double-checked the all references.

Reviewer 2 Report

The manuscript of review article “Exhaustive plant profile of “Dimocarpus longan Lour” with significant phytomedicinal properties: A Literature Based-Review” authored by B. L Kim et al. fully the active components and pharmacological activities of Dimocarpus longan Lour. The comments and suggestions for the manuscript are as the following.

  1. Page 2 Line 49, 51. “4-O-methyl gallic acid” and “4-O-methyl gallic” should be revised to “4-O-methyl gallic acid” and “4-O-methyl gallic”. “O” stands for oxygen atom and must be in italics. Please revised all similar errors.
  2. Page 3 Line 69. “22-36mm” and “6-19g” should be revised to “22-36 mm” and “6-19 g”. A space is required between number and unit. Please revised all similar errors.
  3. Page 4 Line 144-155. “All chemical compounds structures were drawn via the Chem Draw tool.” According to the chemical structure draw of Figure 02 in the manuscript, I think the chemical structure was not drawing by ChemDraw software. On the other hand, all structures and compound names in Figure 02 are not standardized and normalized. The size of graphic structures is inconsistent and distorted. Some compounds don’t show the fully stereochemistry, such as narirutin and proanthocyanidin.
  4. Page 5, Line 193; Page 7, Line 250; Page 14, Line 288; Page 20, Line 569. “Figure 01”, “Figure 02”, “Table 03”, and “Table 04” should be revised as “Figure 1”, “Figure 2”, “Table 3”, and “Table 4”. And please revised all similar errors in the content.
  5. Page 7. In the Table 03, “p-Coumaric” and “hesperidin” should be revised as “p-Coumaric acid” and “Hesperidin”.
  6. Page 12 Line 274. The chemical structure of brevifolin is error. Please check and revised it.
  7. Page 15, Line 292. “D. longan Lour.” should be revised as “ longan Lour.” The scientific name must be italicized. Please revised all similar errors.
  8. Page 18, Line 451-452. “The significant Phar-451 macological Activities of isolated compounds of Longan represented by Table 4.” should be revised as “The significant pharmacological activities of isolated compounds of longan represented by Table 4.”
  9. Page 18, Line 474. “MMP+-induced Neurotoxicity” should be revised as ““MMP+-induced neurotoxicity”.
  10. Page 19, Line 521. “IC50” should be revised as “IC50”.
  11. Please unify the writing of the unit. Please choose one of “mL” or “ml”.
  12. Page 20. In the Table 04, I don’t know why doxorubicin was filled in the compound name & chemical class. In the ref. 37 and 61, doxorubicin just was used the positive control, and it is not isolated from longan Lour. Please delete it.
  13. Page 24. There are many errors that need to be corrected in the reference part. Please follow the guidelines of the journal and be carefully revised all the references.
    • 1: “Lin, Y.; Lai, Z.; Tian, Q.; Lin, L.; Lai, R.; Yang, M.; Zhang, D.; Chen, Y.; Zhang, Z. Endogenous target mimics 613 down-regulate miR160 mediation of ARF10, -16, and -17 cleavage during somatic embryogenesis in 614 Dimocarpus longan Lour. Frontiers in plant science 2015, 6, 956, doi:10.3389/fpls.2015.00956.” should be revised as “Lin, Y.; Lai, Z.; Tian, Q.; Lin, L.; Lai, R.; Yang, M.; Zhang, D.; Chen, Y.; Zhang, Z. Endogenous target mimics down-regulate miR160 mediation of ARF10, -16, and -17 cleavage during somatic embryogenesis in Dimocarpus longan Lour. Front. Plant Sci. 2015, 6, 956, doi:10.3389/fpls.2015.00956.” Please revised all similar errors.
    • 2: “Crane, J.H.; Balerdi, C.F.; Sargent, S.A.; Maguire, I.J.H.S.D.d.F.F.C.E.S., Institute of Food; Agricultural 618 Sciences, U.o.F. Longan growing in the Florida home landscape. Institute of Food and Agricultural Sciences, 619 University of Florida. 2005.” What is the “I.J.H.S.D.d.F.F.C.E.S.” ? This situation can be seen in many references, please check and revised.

Finally, I suggested that the current manuscript “Exhaustive plant profile of “Dimocarpus longan Lour” with significant phytomedicinal properties: A Literature Based-Review” need to be carefully revised before accepting.

Author Response

The manuscript of review article “Exhaustive plant profile of “Dimocarpus longan Lour” with significant phytomedicinal properties: A Literature Based-Review” authored by B. L Kim et al. fully the active components and pharmacological activities of Dimocarpus longan Lour. The comments and suggestions for the manuscript are as the following.

  1. Page 2 Line 49, 51. “4-O-methyl gallic acid” and “4-O-methyl gallic” should be revised to “4-O-methyl gallic acid” and “4-O-methyl gallic”. “O” stands for oxygen atom and must be in italics. Please revised all similar errors.

>>Response: We have corrected it in the page 2, 3, 7, 15 and line 50, 52, 57, 94, 95, 239, 251, 331.  

  1. Page 3 Line 69. “22-36mm” and “6-19g” should be revised to “22-36 mm” and “6-19 g”. A space is required between number and unit. Please revised all similar errors.

>>Response: We have corrected it in the page 3, line 70.

  1. Page 4 Line 144-155. “All chemical compounds structures were drawn via the Chem Draw tool.” According to the chemical structure draw of Figure 02 in the manuscript, I think the chemical structure was not drawing by ChemDraw software. On the other hand, all structures and compound names in Figure 02 are not standardized and normalized. The size of graphic structures is inconsistent and distorted. Some compounds don’t show the fully stereochemistry, such as narirutin and proanthocyanidin.

>>Response: We have redrawn the chemical structure of narirutin and proanthocyanidin, besides we have redrawn some structures where the structures are not normalized.

  1. Page 5, Line 193; Page 7, Line 250; Page 14, Line 288; Page 20, Line 569. “Figure 01”, “Figure 02”, “Table 03”, and “Table 04” should be revised as “Figure 1”, “Figure 2”, “Table 3”, and “Table 4”. And please revised all similar errors in the content.

>>Response: We have revised such issues and solved the updated manuscript.

  1. Page 7. In the Table 03, “p-Coumaric” and “hesperidin” should be revised as “p-Coumaric acid” and “Hesperidin”.

>>Response: We have revised and corrected it in page 7.

  1. Page 12 Line 274. The chemical structure of brevifolin is error. Please check and revised it.

>>Response: We have redrawn in the page 12.

  1. Page 15, Line 292. “D. longan Lour.” should be revised as “longan Lour.” The scientific name must be italicized. Please revised all similar errors.

>>Response: We have revised such problems in the whole manuscript and marked as blue color.

  1. Page 18, Line 451-452. “The significant Phar-451 macological Activities of isolated compounds of Longan represented by Table 4.” should be revised as “The significant pharmacological activities of isolated compounds of longan represented by Table 4.”

>>Response: We have corrected it in the page 17, line 450, 451.

  1. Page 18, Line 474. “MMP+-induced Neurotoxicity” should be revised as ““MMP+-induced neurotoxicity”.

>>Response: We have corrected it in the page 18, line 473.

  1. Page 19, Line 521. “IC50” should be revised as “IC50”.

>>Response: We have corrected it in the page 19, line 520.

  1. Please unify the writing of the unit. Please choose one of “mL” or “ml”.

>>Response: We have revised such problems in the whole manuscript and marked as green color.

  1. Page 20. In the Table 04, I don’t know why doxorubicin was filled in the compound name & chemical class. In the ref. 37 and 61, doxorubicin just was used the positive control, and it is not isolated from longan Lour. Please delete it.

>>Response: We have corrected it.

  1. Page 24. There are many errors that need to be corrected in the reference part. Please follow the guidelines of the journal and be carefully revised all the references.
    • 1: “Lin, Y.; Lai, Z.; Tian, Q.; Lin, L.; Lai, R.; Yang, M.; Zhang, D.; Chen, Y.; Zhang, Z. Endogenous target mimics 613 down-regulate miR160 mediation of ARF10, -16, and -17 cleavage during somatic embryogenesis in 614 Dimocarpus longan Lour. Frontiers in plant science 2015, 6, 956, doi:10.3389/fpls.2015.00956.” should be revised as “Lin, Y.; Lai, Z.; Tian, Q.; Lin, L.; Lai, R.; Yang, M.; Zhang, D.; Chen, Y.; Zhang, Z. Endogenous target mimics down-regulate miR160 mediation of ARF10, -16, and -17 cleavage during somatic embryogenesis in Dimocarpus longan Lour. Front. Plant Sci. 2015, 6, 956, doi:10.3389/fpls.2015.00956.” Please revised all similar errors.
    • 2: “Crane, J.H.; Balerdi, C.F.; Sargent, S.A.; Maguire, I.J.H.S.D.d.F.F.C.E.S., Institute of Food; Agricultural 618 Sciences, U.o.F. Longan growing in the Florida home landscape. Institute of Food and Agricultural Sciences, 619 University of Florida. 2005.” What is the “I.J.H.S.D.d.F.F.C.E.S.” ? This situation can be seen in many references, please check and revised.

          >>Response: We have rechecked all the references and modified all the error refferences.

Round 2

Reviewer 2 Report

The authors have provided acceptable justification and answer the questions put upon the first review. But I have one suggestion that the size of chemical name and structures should be uniformed in the Figure 2.

The manuscript can be published after minor revised

Author Response

The authors have provided acceptable justification and answer the questions put upon the first review. But I have one suggestion that the size of chemical name and structures should be uniformed in the Figure 2.

>>Response: Firstly, we are grateful to the reviewer suggestions. We have uniformed the size of all chemical name and structures and represented under the Figure 2 and also marked as blue color.

This manuscript is a resubmission of an earlier submission. The following is a list of the peer review reports and author responses from that submission.

Round 1

Reviewer 1 Report

This review article entitled "Exhaustive Plant Profile of “Dimocarpus longan Lour” with Significant Phytomedicinal Properties: A Literature Based-Review" by the group of Prof. Kim, described that the chemical components and its pharmacological studies from Dimocarpus longan were organizatied and reviewed. The phytochemitry contain the nutrient components and the isolated components while the pharmacological studies, including anti-proliferative, antioxidative, anticancer, anti-inflammatory, immunomodulatory…etc, were reviewed.  
However, due to the author's negligence, there are a lot of mistakes in the review article that shouldn’t appear. Several important problems and errors in the manuscript are listed below.

  1. In the manuscript, most of the cited references are obviously problematic and incorrect, such as Ref 8~18 (Page 3, line 69, 72, 76, 78, 80, 83), Ref 7 (Page 4, line 101)……….etc. And in the References part (Page 27), the format does not meet the requirements of the journal. The format of the journal should include all author name and the journal name.
  2. In Page 3, line 76-77, “Moreover, doxorubicin is major phenolic compound act towards in vitro cancer cell line and extracted the leaf.” The compound “doxorubicin” has never been isolated and reported from Dimocarpus longan, searching by the Reaxys database. If there is the report, please cite the literature. If not, the “doxorubicin” listing in Keywords may seriously lead to misleading readers.
  3. In Figure 02 (Pages 13-17), the chemical structure of almost all compounds is obviously not related to the name displayed below; while some chemical structures are errors.

According to the above review and some unmentioned errors, I suggest this review article should be rejected in this situation. 

Author Response

This review article entitled "Exhaustive Plant Profile of “Dimocarpus longan Lour” with Significant Phytomedicinal Properties: A Literature Based-Review" by the group of Prof. Kim, described that the chemical components and its pharmacological studies from Dimocarpus longan were organizatied and reviewed. The phytochemitry contain the nutrient components and the isolated components while the pharmacological studies, including anti-proliferative, antioxidative, anticancer, anti-inflammatory, immunomodulatory…etc, were reviewed. 

However, due to the author's negligence, there are a lot of mistakes in the review article that shouldn’t appear. Several important problems and errors in the manuscript are listed below.

>>Response: First and foremost, we'd like to express our heartfelt appreciation for the reviewer's time and effort in reviewing our manuscript.

01) In the manuscript, most of the cited references are obviously problematic and incorrect, such as Ref 8~18 (Page 3, line 69, 72, 76, 78, 80, 83), Ref 7 (Page 4, line 101)……….etc. And in the References part (Page 27), the format does not meet the requirements of the journal. The format of the journal should include all author name and the journal name.

>>Response: We have modified the total references in our edited manuscript according the journal guidelines for reference formatting.  

02) In Page 3, line 76-77, “Moreover, doxorubicin is major phenolic compound act towards in vitro cancer cell line and extracted the leaf.” The compound “doxorubicin” has never been isolated and reported from Dimocarpus longan, searching by the Reaxys database. If there is the report, please cite the literature. If not, the “doxorubicin” listing in Keywords may seriously lead to misleading readers.

>>Response: In our updated manuscript, we have already deleted this sentence for having not enough literature in online based on this information.  

03) In Figure 02 (Pages 13-17), the chemical structure of almost all compounds is obviously not related to the name displayed below; while some chemical structures are errors.

>>Response: We have re-drawn all the chemical structures in our updated manuscript and represented them in the table number 3a (page 10), 3b (page 13), 3c (page 14), 3d (page 15), and 3e (page 16). 

Reviewer 2 Report

Comments on: Exhaustive Plant Profile of “Dimocarpus longan Lour” with Significant Phytomedicinal Properties: A Literature Based-Review.

The comment are reported in the attached file.

Author Response

01) The authors claim to have performed an extensive bibliographic search, consulting both a series of online databases to collect bibliographic data on D. longan, and the references reported by each single paper. Making a review is really a huge job. I am surprised that the bibliography, despite the numerous sources, is composed by 80 citations only. I tried to search new references on Pub Med and I found, in a few minutes, some papers which I report below for example:

>>Response: Firstly, we are grateful to reviewer comments. Primarily, we have missed some of the vital literature on that review topics. However, in our revised manuscript, the total citation is 165.

02) The plant is composed of leaves, flowers, fruits and seeds: perhaps the authors could focus only on one of these parts, describing its components and the biological and pharmacological effects. 

>>Response: Especially, in our review study, our aim is to highlight the potential medicinal compounds from the whole plant parts of longan lour. In the part of introduction, pharmacological activities of our review article, we have broadly represented the specific medicinal activities and their insight mechanisms of these compounds from the total plant parts.  

03) Also, may be, there are one or more substances of the phytocomplex that can be considered specific to the plant, or parts of it, and are responsible of specific beneficial functions. Or a substance present in high quantity to which beneficial effects on health can be attributed. If so it would be appropriate to emphasize and characterize this aspects.

>>Response: In our updated manuscript, we have added more information under beneficial health effects where we have emphasized high quantity substances presented in the plant parts. (page 4, line 89-123)  

04) L.169: "... proper functioning of nerves, and muscles of human" needs a reference.
>> Response: We have added a reference on page 08, line 208, 209.

05) Table 3: it is too packed, I suggest dividing it in relation to the part of the plant considered.

>>Response: In our updated manuscript, we have divided the table 3 to the table number 3a (page 10), 3b (page 13), 3c (page 14), 3d (page 15), and 3e (page 16). 

06) Figure 2: the figure should also be slimmed down, so I would recommend dividing it according to the part of the plant in which it is present. Since they are the diagrammatic representations of the chemical structures shown in table 3, the figure could be subdivided with the same criterion as the table. 

>>Response: We have represented these figures in the updated table number 3a (page 10), 3b (page 13), 3c (page 14), 3d (page 15), and 3e (page 16). 

07) Paragraph 3.6.1. and paragraph 3.6.2.: they should be joined.

>>Response: We have joined the paragraph 3.6.1 and paragraph 3.6.2 as the paragraph 3.6.1 in our modified manuscript (page 16).

08) L. 267-269: The definition of antioxidant is unclear and inaccurate. Antioxidants act by protecting biomacromolecules such as proteins, nucleic acids, etc., not only lipids. So besides being an unclear definition is also incomplete.

>>Response: We have edited the definition of antioxidant in the updated manuscript on the page 17, line 280-297.

09) L. 292: is it "recent"?

>>Response: In our updated manuscript, we have changed it according to the information on the page 19, line 355.

10) Table 4: this table is not mentioned in the text.

>>Response: In our modified manuscript, we have mentioned the table number on the page 16, line 265 and page 21, line 389.

11) Reference paragraph: I don't think the references follow the "Processes" format, anyway many of them lack the name of the Journal.

>>Response: We have modified the total references in our edited manuscript according the journal guidelines for reference formatting.  

12) I am not a native speaker, but I believe that the work should be carefully and entirely revised for English language.

>>Response: We have reviewed and rechecked the English write up of the total manuscript.

Reviewer 3 Report

The article Exhaustive Plant Profile of “Dimocarpus longan Lour” with Significant Phytomedicinal Properties: A Literature Based-Review is interesting, however, some sections of the review are too general and provide too little information about Dimocarpus longan. Anti-proliferative activity, antioxidant, antimicrobial, anti-inflamatory, Immunomodulatory and anti aging activity  section can be improved by adding more studies.

introduction - there are some information which are not covered by any references. please improve the introduction with more information regarding the composition of Longan

176 – Table 1 - the energy value can be on the first column in the table?

185-201 – I would recommend that the names of the chemical compounds be written in lower case

198-205 - try to rephrase the sentence, the phrase seems difficult to understand

210-247 – there are some chemical structure  that are not clear ex. Hesperidin, please check and improve the pictures.

254-265 Are there other studies that support this effect?

276  DPPH

3.6.2. Antioxidant activity and Anticancer Activity -   are there studies that support this effect? maybe some values of antioxidant activity would be interesting

288-291 - Latin names should be written in italics

290 – what DL-P01-SI01 means?

3.6.4. Anti-Inflammatory Properties - it seems too general to me if it is possible you can add more examples

3.6.6. Antifungal Activities - Latin names should be written in italics

334 microorganisms, are

337 probiotc

3.6.8. Anti-aging Activities this section should be merged with Antioxidant activity and Anticancer Activity  remain a separate section

The conclusions should be more adapted to the content of the review

Author Response

01) The article Exhaustive Plant Profile of “Dimocarpus longan Lour” with Significant Phytomedicinal Properties: A Literature Based-Review is interesting, however, some sections of the review are too general and provide too little information about Dimocarpus longan. Anti-proliferative activity, antioxidant, antimicrobial, anti-inflamatory, Immunomodulatory and anti aging activity section can be improved by adding more studies.

>>Response; In our modified manuscript, we added more strong evidences according to the updated literatures for these pharmacological activities.

02) Introduction - there are some information which are not covered by any references. please improve the introduction with more information regarding the composition of Longan

>>Response: We have further re-written the introduction part with several updated citations in our edited manuscript (page 3, line 64-68, line 77-78, line 85-123).

03) 176 – Table 1 - the energy value can be on the first column in the table?

>>Response: In the table 1 of the updated manuscript, we have replaced the energy value in the first column in the table page 8.

04) 185-201 – I would recommend that the names of the chemical compounds be written in lower case.

>>Response: We have edited these chemical compounds name in lower in our edited manuscript on the page (9-10) and line (225-245). 

05) 198-205 - try to rephrase the sentence, the phrase seems difficult to understand

>>Response: We have rephrased the sentence on the page 10, line (242-245).

06) 210-247 – there are some chemical structure that are not clear ex. Hesperidin, please check and improve the pictures.

>>Response: We have re-drawn all the chemical structures in our updated manuscript and represented them in the table number 3a (page 10), 3b (page 13), 3c (page 14), 3d (page 15), and 3e (page 16). 

07) 254-265 Are there other studies that support this effect?

>>Response: In our modified manuscript, we have added more information from the updated literatures for these pharmacological effects.

08) 276  DPPH

>>Response: We have modified this name on the Page 17, line 286.

09) 3.6.2. Antioxidant activity and Anticancer Activity -   are there studies that support this effect? maybe some values of antioxidant activity would be interesting.

>>Response: In our edited manuscript, we have added more update evidence based on these pharmacological activities page 17, line 269-272, line 280-318.

10) 288-291 - Latin names should be written in italics.

>>Response: we have written the microorganism name in italic form in our total updated manuscript.  

11) 290 – what DL-P01-SI01 means?

>>Response: We have edited this information on the page 19, line (347-348).

12) 3.6.4. Anti-Inflammatory Properties - it seems too general to me if it is possible you can add more examples.

>>Response: In our modified manuscript, we have added more updated evidence in the inflammatory properties page 18, line 326-338.

13) 3.6.6. Antifungal Activities - Latin names should be written in italics

>>Response: we have written the microorganism name in italic form in our total updated manuscript. 

14) 334 microorganisms, are

>>Response: We have modified this information on the page 21, line 395.

15) 337 probiotc

>>Response: We have modified this information on the page 21, line 398.

16) 3.6.8. Anti-aging Activities this section should be merged with Antioxidant activity and Anticancer Activity remain a separate section.

>>Response: Other reviewers also suggested us to merge the Anti-proliferative, Antioxidant activity and Anticancer Activity sections, By respecting all the reviewers comments, We have added more updated information on these specific pharmacological activities section for specifying these potential medicinal activities perfectly. Additionally, we added more section: 3.6.8. Neuroprotective Activity (page 22); 3.6.9 Anti-Diabetic Effect and Anti-Hyper glycemic Effect (page 22); 3.6.10 Anti-tyrosinase Properties (page 23); 3.6.11 Miscellaneous Activities (page 24).

17) The conclusions should be more adapted to the content of the review

>>Response: In our updated manuscript, we have newly re-written the Concluding Remarks page 30, line 510-529.

Round 2

Reviewer 1 Report

The revised Manuscript “Exhaustive Plant Profile of “Dimocarpus longan Lour” with Significant Phytomedicinal 2 Properties: A Literature Based-Review “is currently not suitable for this journal. There are two reasons. First, the revised manuscript still contains too many errors and more important some errors shouldn't be made (as described below). Second, the format and drawing of the chemical structures are not unified, and lacks the stereochemistry of all compounds.

In the Tables 3a, 3b, and 3c, many chemical structures are incorrect.

Page 11. The stereochemistry of sugar moiety on isoquercitrin must be assigned because the representative of plane structure refers to many other known compounds, such as hyperoside.

Page 11. The structures of epicatechin, caffeic acid, narirutin, and naringin are incorrect.

Page 12. Rhoifolin, hesperidin, methyl hesperidin, and naringenin belong to “Flavonoids”, not to “Polyphenol”.

Page 12. The structures of rhoifolin, hesperidin, methyl hesperidin, and epicatechin are incorrect.

In the references, there are many errors, such as

  1. the format is not uniform and not routinely used in this journal. For example, the cited journal name should be abbreviated.
  2. Scientific name and journal name are not italic.
  3. Many references repeated. For example, Ref 15/Ref 20, Ref 3/Ref 46, Ref 12/Ref 61, Ref 19/Ref 59, Ref 21/Ref 66, Ref 62/Ref 74, Ref 67/Ref 75, Ref 58/Ref 73, Ref 70/Ref 77, Ref 69/Ref 76, Ref 72/Ref 79, Ref 22/Ref 80, Ref 23/Ref 81………………etc.
  4. Many references are not even assigned the journal name. For example, Ref 15, Ref 43, Ref 50, Ref 51~72, Ref 82, Ref 86, Ref 101, Ref 106, Ref 109~111, Ref 117~128……..etc.

Ref. 4: Pham, V.; Herrero, M.; Hormaza, J.J.A.o.A.B. Fruiting pattern in longan, Dimocarpus longan: from pollination to aril development. Annals of Applied Biology 2016, 169, 357-368.

Pham, V.T.; Herrero, M.; Hormaza, J.I.J.A.o.A.B. Fruiting pattern in longan, Dimocarpus longan: from pollination to aril development. Ann. Appl. Biol. 2016, 169, 357-368.

Ref: 15. Bai, X.; Pan, R.; Li, M.; Li, X.; Zhang, H.J.M. HPLC profile of longan (cv. Shixia) pericarp sourced phenolics and their antioxidant and cytotoxic effects. 2019, 24, 619.

Bai, X.; Pan, R.; Li, M.; Li, X.; Zhang, H.J.M. HPLC profile of longan (cv. Shixia) pericarp sourced phenolics and their antioxidant and cytotoxic effects. Molecules 2019, 24, 619.

Ref: 20. Bai, X.; Pan, R.; Li, M.; Li, X.; Zhang, H. HPLC Profile of Longan (cv. Shixia) Pericarp Sourced Phenolics and Their Antioxidant and Cytotoxic Effects. Molecules (Basel, Switzerland) 2019, 24, doi:10.3390/molecules24030619.

Author Response

The revised Manuscript “Exhaustive Plant Profile of “Dimocarpus longan Lour” with Significant Phytomedicinal 2 Properties: A Literature Based-Review “is currently not suitable for this journal. There are two reasons. First, the revised manuscript still contains too many errors and more important some errors shouldn't be made (as described below). Second, the format and drawing of the chemical structures are not unified, and lacks the stereochemistry of all compounds.

01) In the Tables 3a, 3b, and 3c, many chemical structures are incorrect.

>>Response: We have redrawn all the chemical structures of Tables 3a, 3b, 3c, 3d and 3e.

02) Page 11. The stereochemistry of sugar moiety on isoquercitrin must be assigned because the representative of plane structure refers to many other known compounds, such as hyperoside.

>>Response: The stereochemistry of sugar moiety on isoquercitrin has been changed and redrawn in corrected form on (Page 11, table 3a).

03) Page 11. The structures of epicatechin, caffeic acid, narirutin, and naringin are incorrect. 

>>Response: The structures of epicatechin, caffeic acid, narirutin, and naringin have been redrawn in corrected form. (Page 11 &12, table 3a).

04) Page 12. Rhoifolin, hesperidin, methyl hesperidin, and naringenin belong to “Flavonoids”, not to “Polyphenol”.

>>Response: Rhoifolin, hesperidin, methyl hesperidin, and naringenin are belong to “Flavonoids” that have been mentioned in page 13, table 3a.

05) Page 12. The structures of rhoifolin, hesperidin, methyl hesperidin, and epicatechin are incorrect.

>>Response: The structures of rhoifolin, hesperidin, methyl hesperidin, and epicatechin have been drawn correctly that are mentioned in table 3a, page 11 &13.

 In the references, there are many errors, such as

06) The format is not uniform and not routinely used in this journal. For example, the cited journal name should be abbreviated.

>>Response: We have reformed the total citations and bibliography of our manuscript by EndNote with Journal recommended Citation style (MDPI ACS Journal).  

07) Scientific name and journal name are not italic.

>>Response: We have corrected the total citations.

08) Many references repeated. For example, Ref 15/Ref 20, Ref 3/Ref 46, Ref 12/Ref 61, Ref 19/Ref 59, Ref 21/Ref 66, Ref 62/Ref 74, Ref 67/Ref 75, Ref 58/Ref 73, Ref 70/Ref 77, Ref 69/Ref 76, Ref 72/Ref 79, Ref 22/Ref 80, Ref 23/Ref 81………………etc.

>>Response: We have corrected the errors now.

09) Many references are not even assigned the journal name. For example, Ref 15, Ref 43, Ref 50, Ref 51~72, Ref 82, Ref 86, Ref 101, Ref 106, Ref 109~111, Ref 117~128……..etc.

>>Response: We have corrected all the errors.  

Ref. 4: Pham, V.; Herrero, M.; Hormaza, J.J.A.o.A.B. Fruiting pattern in longan, Dimocarpus longan: from pollination to aril development. Annals of Applied Biology 2016, 169, 357-368.

Pham, V.T.; Herrero, M.; Hormaza, J.I.J.A.o.A.B. Fruiting pattern in longan, Dimocarpus longan: from pollination to aril development. Ann. Appl. Biol. 2016, 169, 357-368.

Ref: 15. Bai, X.; Pan, R.; Li, M.; Li, X.; Zhang, H.J.M. HPLC profile of longan (cv. Shixia) pericarp sourced phenolics and their antioxidant and cytotoxic effects. 2019, 24, 619.

Bai, X.; Pan, R.; Li, M.; Li, X.; Zhang, H.J.M. HPLC profile of longan (cv. Shixia) pericarp sourced phenolics and their antioxidant and cytotoxic effects. Molecules 2019, 24, 619.

Ref: 20. Bai, X.; Pan, R.; Li, M.; Li, X.; Zhang, H. HPLC Profile of Longan (cv. Shixia) Pericarp Sourced Phenolics and Their Antioxidant and Cytotoxic Effects. Molecules (Basel, Switzerland) 2019, 24, doi:10.3390/molecules24030619.

Reviewer 2 Report

The manuscript after the revision has improved considerably, both in terms of organization and content.

There are some details which, in my opinion, should be perfected. Below is the list:

Abstract L. 32: “while” should be deleted.

Introduction L. 61: “Longan” should be “longan”.

  1. 64: Not→ not.
  2. 68: After diameter add comma.
  3. 90: delete “of”.
  4. 201: Instead of “Longan Tree” it would be better “Dimocarpus longan”.

L 202: Replace “Nutrient Components” with “Nutrient Components of the fruit”.

  1. 257: Replace “Pharmacological Activities” with “Pharmacological Activities of Dimocarpus longan”.
  2. 258: I would change the order of the paragraphs in this way:

3.6.1. Anti-proliferative, Antioxidant activity and Anticancer Activity

3.6.2. Anti-Inflammatory Properties

3.6.3. Immunomodulatory Activities

3.6.4. Prebiotic Activities

3.6.5. Anti-microbial Activities

3.6.6. Anti-fungal Activities

3.6.7. Neuroprotective Activity:

3.6.8. Anti-aging Activities

3.6.9 Anti-Diabetic Effect and Anti-Hyper glycemic Effect:

3.6.10 Anti-tyrosinase Properties:

3.6.11 Miscellaneous Activities:

  1. 453: 3.6.10 “Anti-tyrosinase Properties” subparagraph. I should add few information about the role of this enzyme in oxidation of phenolic compounds found in fruits and vegetables, and why the inhibition of tyrosinase is important.
  2. 517-518: Quotation marks are not needed if italics are used.

Check throughout the manuscript if “Dimocarpus longan” is written appropriately.

Author Response

The manuscript after the revision has improved considerably, both in terms of organization and content.

There are some details which, in my opinion, should be perfected. Below is the list:

01) Abstract L. 32: “while” should be deleted.

>>Response: We have changed the world in our revised manuscript in page 02, line 35.  

02) Introduction L. 61: “Longan” should be “longan”.

>>Response: we have modified it to longan in the page page 03, line 63

03) 64: Not→ not

>>Response: we have changed it to not in page 03, line 67

04) 68: After diameter add comma

>>Response: we have added a comma after diameter in page 03, line 71

05) 90: delete “of”

>>Response: we have deleted the preposition from our manuscript in page 04, line 93

06) 201: Instead of “Longan Tree” it would be better “Dimocarpus longan”

>>Response: We have modified the “Longan Tree” to “Dimocarpus longan” in page 08, line 202    

07) L 202: Replace “Nutrient Components” with “Nutrient Components of the fruit”.

>>Response: We have replaced the “Nutrient Components” with “Nutrient Components of the fruit” on page 08, line 203 

08) 257: Replace “Pharmacological Activities” with “Pharmacological Activities of Dimocarpus longan”

>>Response: we have modified “Pharmacological Activities” to “Pharmacological Activities of Dimocarpus longan” on page 21, line 259   

09) 258: I would change the order of the paragraphs in this way:

>>Response: We have modified the order of these paragraph according to the reviewer suggestions

10) 453: 3.6.10 “Anti-tyrosinase Properties” subparagraph. I should add few information about the role of this enzyme in oxidation of phenolic compounds found in fruits and vegetables, and why the inhibition of tyrosinase is important

>>Response: We have added some information on the paragraph of “Anti-tyrosinase Properties” about the role of this enzymes in oxidation of phenolic compounds founds in fruits and vegetables, and why the inhibition of tyrosinase is important on page 28, line (455-462).   

11) 517-518: Quotation marks are not needed if italics are used.

>>Response: We have removed the quotation and made it italics form on page 35, line (526, 527, 532)

12) Check throughout the manuscript if “Dimocarpus longan” is written appropriately

>>Response: We have thoroughly checked the total manuscript and corrected this appropriate plant name (Dimocarpus longan) in some portion in page 03, line 80; page 04, line 96; page 05, line (131, 135)   

Reviewer 3 Report

 Significant improvements have been made to the article, but there are still many mistakes in the manuscript., such as:

  1. Table 3a,b and c contains many wrong chemical formulas (caffeic acid, naringin, epicatechin, methyl hesperidin etc) which need to be corrected
  2. Naringin, hesperidin, naringenin are flavonoids
  3. The bibliography is not arranged according to the journal guide 

Authors must carefully check and correct all formulas and bibliography.

Author Response

Significant improvements have been made to the article, but there are still many mistakes in the manuscript., such as:

  • Table 3a,b and c contains many wrong chemical formulas (caffeic acid, naringin, epicatechin, methyl hesperidin etc) which need to be corrected

>>Response: We have redrawn all the chemical structures and represented in the table no 3a, 3b, 3c, 3d and 3e.

  • Naringin, hesperidin, naringenin are flavonoids

>>Response: We have corrected it

  • The bibliography is not arranged according to the journal guide. Authors must carefully check and correct all formulas and bibliography.

>> Response: We have reformed the total citations and bibliography of our manuscript by EndNote with Journal recommended Citation style (MDPI ACS Journal).